# The Role of Sulphate and Phosphate Ions in the Recovery of Benzoic Acid Self-Enhanced Ozonation in Water Containing Bromides

**DOI:** 10.3390/molecules26092701

**Published:** 2021-05-05

**Authors:** Lilla Fijołek, Joanna Świetlik, Marcin Frankowski

**Affiliations:** Faculty of Chemistry, Adam Mickiewicz University, Uniwersytetu Poznańskiego 8, 61-614 Poznań, Poland; askas@amu.edu.pl

**Keywords:** ozonation, bromides, sulphate, phosphates, radicals, advanced oxidation processes

## Abstract

The ozonation of aromatic compounds in low-pH water is ineffective. In an acidic environment, the decomposition of ozone into hydroxyl radicals is limited and insufficient for the degradation of organic pollutants. Radical processes are also strongly inhibited by halogen ions present in the reaction medium, especially at low pH. It was shown that even under such unfavorable conditions, some compounds can initiate radical chain reactions leading to the formation of hydroxyl radicals, thus accelerating the ozonation process, which is referred to as so-called “self-enhanced ozonation”. This paper presents the effect of bromides on “self-enhanced ozonation” of benzoic acid (BA) at pH 2.5. It is the first report to fully and quantitatively describe this process. The presence of only 15 µM bromides in water inhibits ozone decomposition and completely blocks BA degradation. However, the effectiveness of this process can be regained by ozonation in the presence of phosphates or sulphate. The addition of these inorganic salts to the bromide-containing solution helps to recover ozone decomposition and BA degradation efficiency. As part of this research, the fractions of hydroxyl, sulphate and phosphate radicals reacting with benzoic acid and bromides were calculated.

## 1. Introduction

Many industries, for example, the production of ammunition, pharmaceuticals, mining, the steel industry, and the galvanizing and phosphoric industry, generate highly acidic sewage, which may contain many undesirable and toxic impurities. As a consequence, there is a need to develop effective purification methods before discharge into the environment. If the oxidation process of such water leads to the formation of hydroxyl radicals, then the destruction of organic compounds present in the water can be expected. Therefore, the application of an advanced oxidation process (AOP) would be helpful to increase the purification efficiency. However, most AOPs carried out at low pH are not very effective because of the high chemical stability of most oxidants in the acidic environment; the one exception is the Fenton process (and its modifications). Therefore, the generation of hydroxyl radicals in an acidic environment is of great interest. One of the AOPs that may be used to purify such water is the ozonation process. Stability of ozone molecules is strongly pH dependent. The lower the pH, the more stable the ozone molecule is. However, the oxidation potential of the ozone decomposition product, the hydroxyl radical, is much higher, and therefore ^•^OH is responsible for oxidation of most organic contaminants. This implies that, at low pH, the ozone molecule is relatively stable and is not very reactive towards most organic water pollutants [1,2]. The ability of some aromatic compounds to begin radical chain reactions destroying ozone was noticed a long time ago [3]. Pi et al. [4] described the reaction of p-chlorobenzoic acid that led to the formation of radicals. At first, during ozonation of an aromatic molecule, a double bond is created and H_2_O_2_ is generated [4]. Afterwards, the dissociation of hydrogen peroxide leads to the formation of HO_2_^−^ ions that can subsequently initiate radical chain reactions and, finally, •OH radical formation. However, such a reaction path requires relatively high pH to produce enough HO_2_¯ ions to initiate ozone decomposition. Buffle and Von Gunten [5] showed the ability of phenols and amines to begin chain reactions generating hydroxyl radicals. This works typically for AOPs carried out at circumneutral pHs. Recently, Huang et al. [6] showed that ozonation of benzoic acid (BA) at low pH (2.3) started a self-enhanced radical degradation chain reaction that led to effective removal of acid. The authors [6] suggested that the intermediate products of BA ozonation enhance •OH production in the system and contribute to its degradation and proposed the following possible reactions [6]:

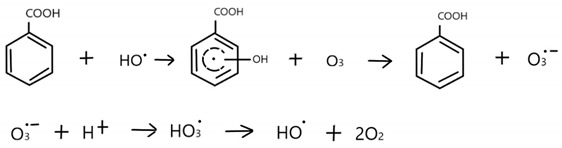



Huang et al. [6] also suggested that, during aromatic ring opening, HO^•^_2_ can be formed and subsequently, in reaction with ozone molecules, hydroxyl radicals are generated.

The process can be considered as an autocatalytic reaction since the higher the concentration of organic substrate, the higher the degree of destruction observed. It should be emphasized that this study [6] was conducted on deionized model water. However natural waters and waste waters contain mineral compounds that can significantly affect the course and effectiveness of the self-enhanced ozonation process. Among these constituents are halide ions, which decrease the efficiency of radical processes, especially at low pH [7,8]. This is important because both chlorides and bromides occur naturally in water. The chloride content in water ranges from a few to several hundred mg/L. The concentration of bromide ions in fresh water ranges from 10 μg/L to about 1000 μg/L, and in marine water the content is much higher. In many coastal areas, bromide concentrations may be higher due to the influence of seawater [9]. Due to the presence of halogens, the treatment of saline water and wastewater is challenging and there is an increasing need for techniques removing organic contaminants from brackish industrial and municipal waste water [10]. Chlorides influence chain radical reactions, reducing the effectiveness of advanced oxidation processes [11,12,13,14,15,16,17]. It has also been shown that the process of self-enhanced ozonation is strongly inhibited by the presence of chloride ions in the solution [7]. Similarly to chlorides, this process should be inhibited by the presence of bromides. According to Grebel et al. [8], “Br^−^ is potentially of more significant concern for AOPs. Despite occurring at a 675-fold lower concentration than Cl^−^, Br^−^ is the most important scavenger of HO^•^ in seawater, removing up to ~93%”. Conducting the ozonation process at low pH limits the formation of bromates [18]; however, less reactive forms of bromine (bromide and dibromine radicals) are generated [8]. It follows that AOPs in the presence of halogen ions are not very effective. This is the reason that treatment of such water/wastewater is inefficient. However, recently we showed [7] that conducting of radical processes in the presence of chlorides is possible. This was achieved by the addition of phosphates to ozonated water. Phosphates can generate radicals, which contribute to the re-decomposition of ozone and the increased degradation and mineralization of pollutants. Considering that chlorides react in a very similar way to bromides, we expect that results observed for chlorides should be also applicable in the case of bromides. Similar to phosphate, sulphate can generate active radicals as well. Therefore, this study aimed not only to quantify the effect of the presence of bromide on self-enhanced ozonation, but also to check whether and how phosphates and sulphates can counterbalance the influence of bromides.

Benzoic acid has been detected in wastewater from the wood production industry and in foundry waste leachates, as well as in extracts of fly ash from municipal incinerators. BA is increasingly used in the production of diethylene, dipropylene glycol, dibenzoate, plasticizers, adhesive formulations, paints and coatings, and secondary oil production. Benzoic acid and sodium benzoate are used as preservatives in beverages, fruit products, chemically leavened baked goods, and condiments [19]. Its widespread utilization makes BA an environmentally significant anthropogenic pollutant. At the same time, benzoic acid is often used as a model compound in AOPs; therefore, the rate constants for its reaction with various oxidizingspecies are known. The self-enhanced ozonation reaction has also been carried out at low pH and described for BA. Thus, in this study, benzoic acid was chosen as the model compound.

## 2. Results and Discussion

### 2.1. Ozone Decomposition

Under the experimental conditions of this work (pH 2.5), about 20% of the initial amount of ozone decomposes spontaneously within an hour. The introduction of 15 µM of bromides has practically no effect on this process (Figure 1). Degradation of organic water pollutants by means of molecular ozone is ineffective. Since only the reaction with ozone decay products (hydroxyl radicals) is effective, the more that ozone dissolved in water decomposes with the formation of hydroxyl radicals, the higher the degradation degree of the organic compound can be expected. Introducing benzoic acid into the water saturated with ozone initiates the process of its decomposition towards hydroxyl radicals (Huang et al. 2015). If BA is introduced into bromide-free water, ozone can completely decompose within 50 min (Figure 1). 

The introduction of bromides to water containing ozone and BA decreased the amount of ozone decomposed (Figure 1). Huang et al. [6] stated that benzoic acid was degraded efficiently by ozone via a •OH-induced mechanism under acidic conditions and the intermediates produced during BA degradation are involved in ozone activation to generate •OH. Bromides scavenge the hydroxyl radicals from the solution and hence quench the radical chain reactions, leading to ozone decomposition. An increase of bromide from 1 to 12.5 µM can slow the ozone decomposition process down to about 50% within an hour. A further increase of bromide to 15 µM does not contribute to a further inhibition of ozone decomposition in the presence of BA (Figure 1). On the basis of our experience [7], we fit kinetic models to the results obtained in this work. We found that the reactions of ozone decomposition (Figure 1) and BA degradation in the presence of bromides are pseudo-second order reactions. At the same time, with the increase in the amount of bromide in the solution, a decrease in the rate constant of ozone decomposition was observed (Figure 2).

### 2.2. BA Degradation

As shown in Section 2.1, micromolar amounts of bromides inhibit the ozone decomposition in the presence of BA molecules. In the reaction with bromides, the amount of •OH radicals is reduced, which should also result in a reduction in the efficiency of BA degradation. In pure bromide free water, BA is completely degraded within 10 min (Figure 2). Only 1 µM of bromides in the solution decreases the efficiency of the process in the first minutes and the increased amount of bromides (up to 10 µM) contributes to a gradual inhibition of the BA degradation. It has also been observed that the further increase of bromide concentration (from 10 µM to 15 µM) does not significantly decrease the process efficiency (Figure 3), which is related to the amount of ozone decomposed (Figure 1). Therefore, we did not proceed with experiments using bromide concentrations above 15 µM.

At the same time, with the increasing amount of bromides, a decrease in the BA degradation reaction rate constant was observed (Figure 4).

Hydroxyl radicals react with benzoic acid with a rate constant of *k* = 5.6 × 10^9^ M ^−1^s^−1^ [20] and with bromide *k* = 1 × 10^10^ M ^−1^s^−1^ [20], which seems to explain the decrease in the efficiency of BA ozonation in the presence of bromides. 

### 2.3. Recovery of the Radical Process at Low pH

Bromides strongly inhibit the efficiency of radical processes [8]. However, as it was shown [7], the ozonation process of aromatic compounds in the presence of halogens, at low pH, is possible when phosphates are introduced to the solution. We decided to check whether the solution that worked for chlorides would be applicable in the case of bromides. Ozone decomposition experiments were started from initial concentrations of 24 µM of BA and 15 µM of bromides. When only 24 µM of BA was added to water (pH 2.5) saturated with ozone, the latter decomposed within about 50 min (Figure 5). When, in the next step, 15 µM of bromides were added to the system, the amount of ozone decomposed decreased from 100% to approx. 45% (60 min; Figure 5). Introduction of an increasing amount of phosphate into the ozonated solution containing benzoic acid (24 µM) and bromides (15 µM) resulted in a gradual increase in the degree of ozone decomposed (Figure 5).

In the presence of 50 mM of phosphates and bromides (15 µM), ozone decomposes completely within 60 min (Figure 5). As mentioned above, the more ozone dissolved in water decomposes with the formation of hydroxyl radicals, the higher the degradation degree of the organic compound can be expected. As it was shown earlier [7], the NB/BA degradation in the presence of phosphates proceeds via a radical pathway, even in the presence of chlorides. It can therefore be expected that when the concentration of phosphates in the water containing bromide increases, the efficiency of BA degradation should also increase. Indeed, such results were observed. The introduction of phosphates (10 mM) accelerated the decomposition of BA (Figure 6). Increasing the phosphate concentration to 25 mM resulted in a further increase in BA removal efficiency to 60%. In addition, it was observed that further increasing the concentration of phosphates to 50 mM did not cause a significant increase in benzoic acid degradation (Figure 6), even though ozone was decomposed to a higher degree (Figure 5).

The differences in the reaction rate constants partially explain the observed results. As indicated above, the rate constant of the hydroxyl radical reaction with bromides is an order of magnitude higher than with BA; therefore, ^•^OH will react faster with bromides than BA. This implies that the bromides will be oxidized at first and since hydroxyl radicals are used in this process, and a decrease in ozonation efficiency of BA will also be observed. The introduction of phosphates into the water contributes to the recovery of BA degradation. The observed phenomenon is related to the fact that phosphates in reaction with the hydroxyl radical generate phosphate radicals [7,21], which exhibit oxidizing properties. Similarly to phosphates, sulphates in reaction with ^•^OH also generate radicals that show oxidizing properties [22,23], which is discussed below. In the present study, when sulphates were introduced into water containing 15 µM of bromides, the recovery of ozone decomposition was observed (Figure 7). This is an important observation from an applicational point of view. It should be noticed that in practical/real conditions, to increase the efficiency of the ozonation process in the presence of chlorides/bromides, it is necessary to introduce phosphates into the water, while sulphates are usually already present in a large amount in both natural and postprocess waters and sewage.

When BA is introduced into ozone-saturated water, ozone decomposes completely within 50 min. The introduction of 15 µM of bromide into the system causes a decrease in the amount of ozone decomposed (Figure 7). Similarly to phosphates, the introduction of sulphates into the solution raises the degree of ozone decomposition (Figure 7). However, the application of sulphates has been shown to be less effective than phosphates, since a slightly lower efficiency of BA degradation recovery is observed (Figure 8).

Introduction of phosphates contributes to the recovery of ozone decomposition and degradation of BA. Phosphates react with hydroxyl radicals to form phosphate radicals that exhibit oxidizing properties [24,25,26]. Since phosphate radicals have oxidizing properties, their appropriate excess over bromides contributes to the effective degradation of BA in a similar way to chlorides [7]. Other ions, such as perchlorate, which do not generate radicals, do not contribute to the degradation of pollutants [7]. Therefore, the reaction of phosphates with hydroxyl radicals should determine the course of the process in the presence of halogen ions. As was stated above, sulphates also react with •OH in an acidic environment, generating active radicals that have oxidizing properties [22,23].

Taking into account the reaction rate constants, the amount of sulphate radicals produced in the solution should be comparable to the amount of phosphate radicals (the magnitudes of the reaction rate constants are of the order 10^6^) [21,22,23]:^●^OH + H_2_PO_4_^−^ → ^−^OH + H_2_PO_4_^●^   *k* = 2.2 × 10^6^ M ^−1^s^−1^ **(1)
^●^OH + SO_4_^2−^ → SO_4_^●−^ + OH^−^   *k* = 1.18 × 10^6^ M ^−1^s^−1^(2)
** Some sources give a value: *k* = 2 × 10^4^ M ^−1^s^−1^ [27,28]. In the calculations, the value of k given in the reaction was adopted.

In addition, the reaction rate constant of BA with sulphate radicals is an order of magnitude higher than that with phosphate radicals [29,30,31]:H_2_PO_4_^●^ + BA → products   *k* = 2.8 × 10^8^ M^−1^ s^−1^(3)
SO_4_^●−^ + BA → products   *k* = 1.2 × 10^9^ M^−1^ s^−1^(4)

Therefore, it can be expected that the recovery of radical BA degradation by sulphates should be higher than by phosphates. However, the obtained results show that processes in the presence of phosphates are more effective (Figure 8), despite the fact that the rate constant of benzoic acid oxidation by phosphate radicals is an order of magnitude lower than the rate constant for the oxidation of BA by sulphate radicals. Nevertheless, both sulphate and phosphate radicals, similarly to hydroxyl radicals, also react with bromides, which may explain the obtained results. The constant rates of those reactions are [32]:H_2_PO_4_^●^ + Br^−^ → products   *k* = 8.0 × 10^8^ M^−1^ s^−1^ (pH = 4)(5)
SO_4_^●−^ + Br^−^ → products   *k* = 3.5 × 10^9^ M^−1^ s^−1^ (pH = 7)(6)

A comparison of those rate constants indicates that sulphate radicals are more strongly quenched by bromides, which can explain the lower recovery efficiency of radical degradation by sulphates compared to phosphates. To verify the obtained results, the fraction of hydroxyl radicals reacted with BA salts and bromides was determined and the results are presented in Table 1. The used calculation methods are shown in the calculations section.

The obtained results confirmed the above assumptions. Processes in the presence of sulphates are more strongly quenched by bromides. During ozonation of BA in a solution containing bromide and phosphates, 37.4% of ^•^OH is scavenged by the bromides, while in the presence of sulphate and bromides the amount of scavenged radicals is higher and equal to 44.4% (Table 1). At the same time, the fraction of hydroxyl radicals reacting with benzoic acid is lower in the presence of phosphates than sulphates (Table 1).

According to Huang et al. [6], hydroxyl radicals are formed during BA degradation and hydroxylated intermediates (HBA) are chain carriers. In this reaction, 1 mole of BA consumes 4 moles of ozone. Assuming that 1 mole of ^•^OH is generated per 1 mole of ozone decomposed and that 1 mole of BA leads to the decomposition of 4 moles of ozone, then, in the case of this work, from 120 µM of initial ozone amount, 96 µM (80% of 120 µM) of ^•^OH could form. According to the data obtained in the tested systems, 20% of the initial ozone amount undergoes self-decomposition (Figure 1) and studies have shown that tert-butyl alcohol (TBA-hydroxyl radical scavenger) has no effect on this process, which means that this 20% of ozone decomposes without ^•^OH formation. It was also observed that the addition of TBA to the ozonated BA solution lowers the acid degradation efficiency to 20% (Figure 3). Additionally, the recovery of the reaction in the presence of bromides and phosphates is inhibited by TBA to 20% (Figure 6), which confirms the correctness of the above assumption. Considering the obtained values (Table 1), during ozonation in the presence of bromide, 28.9% of hydroxyl radicals might react with phosphates, which can lead to the formation of 27.7 µM H_2_PO_4_^●^ (based on Equation (2)). This is almost twice as much as the amount of bromides in solution. In the case of sulphates, only 15.9% of hydroxyl radicals are involved in the reaction, which translates in the possibility of the formation of 15.3 µM SO_4_^●−^. The calculations indicate that the amounts of sulphate radicals generated in the solutions are lower than the amounts of phosphate radicals in the systems in which phosphates were applied. Phosphate and sulphate radicals, like hydroxyl radicals, can also oxidize BA in solution. Therefore, the higher efficiency of the ozonation process in the presence of phosphates can be attributed to the greater amount of generated phosphate radicals. Fractions of sulphate and phosphate radicals reacting with BA bromide and its recombination are shown in Table 2. The calculations are based on Equations (22)–(24).

The analysis of the obtained results shows that only a small proportion of phosphate radicals (9.1%) can oxidize BA, while as much as 33.1% of SO_4_^●−^ can take part in the reaction (Table 2). As a consequence, a higher efficiency of BA degradation by sulphate radicals should be observed. The results obtained in the experiments, however, show that a much higher efficiency of BA removal is noticed in the system where phosphates were added (Figure 8). This is most likely related to the previously described lower amount of sulphate radicals generated in the reaction system and their lower oxidizing potential (2.437 ± 0.019 V) compared to H_2_PO_4_^●^ radicals (2.75 ± 0.01 V) [33]. Additionally, it should be emphasized that during ozonation in the presence of sulphates, the fraction of hydroxyl radicals reacting with bromide is greater, which is why this process is inhibited to a greater extent (Table 1). Moreover, sulphate radicals are scavenged to a higher extent by bromides than phosphate radicals (Table 2). More than 60% of the sulphate radicals generated can react with bromides and oxidize them to an unstable bromine radical (Br^•^) [34], which is a much weaker oxidant than SO_4_^●−^/H_2_PO_4_^●^ radicals. Br^•^ are characterized by an oxidizing potential of 1.96 ± 0.02 V [33] and in an acidic environment they easily form even less reactive Br^•−^ (1.63 ± 0.02 V) [33]. 

Taking into account the reactions given in the calculation section, 74.7% of the H_2_PO_4_^●^ radicals (Table 2) may undergo secondary reactions with the formation of H_2_P_2_O_8_. This peroxide is also a strong oxidant and can oxidize bromides in acidic conditions [35]. As presented in Table 1, the amount of BA oxidized by hydroxyl radicals in the presence of phosphates is lower than in the presence of sulphates. Hence, the higher efficiency of the recovery process in the presence of bromides and phosphates can be explained by the action of H_2_PO_4_^●^ radicals (Table 2), their generation in a larger amount than sulphate radicals, and the possibility of their recombination with bromide-oxidizing H_2_P_2_O_8_ peroxide. On the other hand, in the case of sulphate radicals, the process may proceed by a direct reaction of SO_4_^●−^ with bromides. Both phosphate and sulphate radicals are formed in the reaction with the hydroxyl radical; therefore, in the presence of TBA, the processes involving these radicals are inhibited (Figure 6). 

## 3. Materials and Methods

The following chemicals were used: benzoic acid (BA) (Sigma-Aldrich, Poznań, Poland, CS reagent, min 99.5%), potassium indigotrisulfonate (Sigma Aldrich, Poznań, Poland, >60%), tert-butyl alcohol (TBA) (Sigma-Aldrich, Poznań, Poland, min. 99.5%), sodium phosphate monobasic (NaH_2_PO_4_) (P) (Acros Organics, Argenta, Poznań, Poland,) 99%), sodium sulfate (Na_2_SO_4_) (S) (Sigma Aldrich, Poznań, Poland, 99%), sodium sulfite (Na_2_SO_3_) (Sigma Aldrich, Poznań, Poland, pure p.a), sodium bromide (NaBr) (Sigma Aldrich, Poznań, Poland, 99.5%). All reagents were used as received.

The aqueous ozone decomposition and ozonation processes were carried out at room temperature for 60 min. For this purpose, to satisfy the reaction vessel ozone demand, before each series of experiments, high purity water (Millipore) with a pH of 2.5 was introduced into the reactor and pre‑saturated with ozone. The pH was adjusted to pH 2.5 with HClO_4_, which did not have an influence on the studied processes. In each experiment, water was saturated by a stream of ozone for 20 min. During ozone decomposition, the amount in water was measured by the indigo method [36]. In all experiments, the average initial ozone concentrations were 120 ± 5 µM. The amounts of benzoic acid in the samples were quantified chromatographically, using a Symmetry C18 column (75 × 4.6 mm, 3.5 mm packing). As a mobile phase, water/acetonitrile was used (50:50); water was acidified to pH 3.0 by H_3_PO_4_. The flow rate of 0.6 mL/min was used. For chromatographic analyses, immediately after sampling, Na_2_SO_3_ (68 mM) was added to quench ozone in the samples. Initial and residual amounts of ozone were also measured by the indigo method. Each experiment was repeated at least three times. The RSD of measurements was lower than 3%. 

### Calculations

To explain the observed phenomena, the fraction of hydroxyl radicals reacting with the reactants (benzoic acid, phosphates or sulphate and bromides) present in the system was determined. The following reactions were included [20]:^●^OH + BA → products   *k* = 5.6 × 10^9^ M ^−1^s^−1^,(7)
^●^OH + Br^−^ → products   *k* = 1 × 10^10^ M ^−1^s^−1^,(8)
and: -in the case of phosphates [21,37]:
^●^OH + H_2_PO_4_^−^ → ^−^OH + H_2_PO_4_^●^   *k* = 2.2 × 10^6^ M ^−1^s^−1^(9)
^●^OH + H_3_PO_4_ → H_2_O + H_2_PO_4_^●^   *k* = 2.6 × 10^6^ M ^−1^s^−1^(10)

-sulphates [22,38]:

^●^OH + HSO_4_^−^ → SO_4_^●−^ + H_2_   *k* = 7.4 × 10^5^ M ^−1^s^−1^(11)

^●^OH + SO_4_^2−^ → SO_4_^●−^ + OH^−^   *k* = 1.18 × 10^6^ M ^−1^s^−1^(12)

The fraction of hydroxyl radicals reacting in the studied systems was determined similarly to the previous calculations [7,39] from the formulas:fraction of ^•^OH reacted with benzoic acid (∫_OH-BA_):
(13)$$\int_{OH-BA}=\frac{k_{OH-BA\;\;}\left[BA\right]}{k_{OH-BA\;\;}\left[BA\right]+k_{OH-P/S\;\;}\left[P/S\right]+k_{OH-Br\;\;}\left[Br\right]}\;•\;100\%\;$$

fraction of ^•^OH reacted with phosphate or sulphate (∫_OH-P/S_:):

(14)$$\int_{OH-P/S}=\frac{k_{OH-P/S\;\;}\left[P/S\right]}{k_{OH-P/S\;}\left[P/S\right]+k_{OH-BA\;}\left[BA\right]+k_{OH-Br\;\;}\left[Br\right]}\;•\;100\%\;$$

fraction of ^•^OH reacted with bromide (∫_OH-Br_:):
(15)$$\int_{OH-Br}=\frac{k_{OH-Br\;}\left[Br\right]}{k_{OH-Br\;\;}\left[Br\right]+k_{OH-BA\;}\left[BA\right]+k_{OH-P/S\;}\left[P/S\right]}\;•\;100\%\;$$
where in Equations (13)–(15):

*k_OH-BA_*—reaction rate constant of *BA* with hydroxyl radical [M s^−1^] 

[*BA*]—concentration of *BA* = 24 × 10^−6^ [M]

*k_OH-P/S_*—reaction rate constant of phosphate/sulphate with hydroxyl radical [M s^−1^]

[*P/S*]—oncentration of phosphates or sulphates = 50 × 10^−3^ [M]

*k_OH-Br-_*—reaction rate constant of bromide with hydroxyl radical 5.6 × 10^9^ [M s^−1^]

[*Br*]—concentration of bromides 15 × 10^−6^ [M]

The calculations consider the dissociation of acids at pH 2.5. The following forms are present in solution: in the case of phosphoric acid: 70.58% H_2_PO_4_^−^ + 29.42% H_3_PO_4_ and sulfuric acid: 76.39% SO_4_^2−^ + 23.61% HSO_4_^−^.

To determine the fraction of phosphate/sulphate radicals reacted in solution, the following reaction was taken into consideration: -for phosphate radicals [29,32]:
H_2_PO_4_^●^ + BA → products   *k* = 2.8 × 10^8^ M^−1^ s^−1^(16)
H_2_PO_4_^●^ + Br^−^ → products   *k* = 8.0 × 10^8^ M^−1^ s^−1^ (pH = 4)(17)
H_2_PO_4_^●^ + H_2_PO_4_^●^ → H_2_P_2_O_8_ + 2H^+^   *k* = 2 × 10^9^ M^−1^ s^−1^ (pH = 3.5) (18)

-for sulphate radicals [29,30,31,32]:

SO_4_^●−^ + BA → products   *k* = 1.2 × 10^9^ M^−1^ s^−1^(19)

SO_4_^●−^ + Br^−^ → products   *k* = 3.5 × 10^9^ M^−1^ s^−1^ (pH = 7) (20)

SO_4_^●−^ + SO_4_^●−^ → S_2_O_8_^2−^ + 2H   *k* = 3.8 × 10^8^ M^−1^ s^−1^ (pH = 3.5) (21)

To determine the fraction of phosphate or sulphate radicals reacting in the tested systems, the following formulas were used:fraction of phosphate/sulphate radicals reacted with benzoic acid (∫_P/S-BA_):
(22)$$\int_{S/P-BA}=\frac{k_{S/P-BA\;\;}\left[BA\right]}{k_{S/P-BA\;}\left[BA\right]+k_{S/P-Br\;}\left[Br\right]+k_{S/P-S/P\;}\left[P/S\right]}\;•\;100\%\;$$

fraction of phosphate/sulphate radicals reacted with bromide (∫_P/S-Br_):

(23)$$\int_{S/P-Br}=\frac{k_{S/P-Br\;\;}\left[Br\right]}{k_{S/P-Br\;\;}\left[Br\right]+k_{S/P-BA\;\;}\left[BA\right]+k_{S/P-OH\;}\left[P/S\right]}\;•\;100\%\;$$

fraction of phosphate/sulphate radical recombination (∫_P/S-S/P_):
(24)$$\int_{S/P-S/P}=\frac{\;k_{S/P-S/P\;\;\;}\left[S/P\right]}{k_{S/P-S/P\;\;}\left[S/P\right]+k_{S/P-Br\;\;}\left[Br\right]+k_{S/P-BA\;\;}\left[BA\right]}\;•\;100\%$$
where in Equations (22)–(24):

*k_S/P-BA_*—reaction rate constant of *BA* with sulphate or phosphate radicals [M s^−1^]

[*BA*]—concentration of *BA* = 24 × 10^−6^ [M]

*k_S/P-Br_*—reaction rate constant of bromides with sulphate or phosphate radicals [M s^−1^]

[*Br*]—concentration of bromides 15 × 10^−6^ [M]

*k_S/P-S/P_*—sulphate/phosphate radical recombination reaction rate constant [M s^−1^]

[*S/P*]—concentration of sulphate (15.3 × 10^−6^) or phosphate radicals (27.7 × 10^−6^) [M]

## 4. Conclusions

Bromides strongly inhibit the process of self-enhanced ozonation. The research showed a quantitative relationship between the concentration of bromides in the solution and the effectiveness of benzoic acid removal. It was shown that the addition of only 15 µM of bromides completely inhibits the BA degradation process. However, the introduction of phosphates into ozonated water containing bromides contributes to recovery of the reaction and increases the efficiency of BA oxidation. This paper shows, for the first time, that the process of radical degradation of pollutants in the presence of a halogen ion is also possible by introducing sulphates into ozonated water. Such a solution opens the possibility of industrial application of the results presented in the paper. Bromides inhibit the ozonation process more strongly in the presence of sulphates than in the presence of phosphates.Phosphates and sulphates in ozonated water may form radicals capable of oxidizing organic pollutants, and therefore their appropriate excess over bromides can contribute to increasing the efficiency of benzoic acid degradation.The recovery of benzoic acid ozonation in the presence of bromides occurs most likely as a result of secondary reactions with phosphate and sulphate radicals. Calculations showed that the efficiency of benzoic acid oxidation by phosphate radicals is lower than by sulphate radicals, however, phosphate radicals are generated to a greater extent.The discussed recovery processes are initiated by hydroxyl radicals, which was confirmed by their inhibition in the presence of TBA. The results obtained in the study indicate that carrying out radical processes in the presence of bromides is possible. This implies that using the proposed solution in the treatment of water and waste water, regardless of the pH and salinity, is possible.

## Figures and Tables

**Figure 1 molecules-26-02701-f001:**
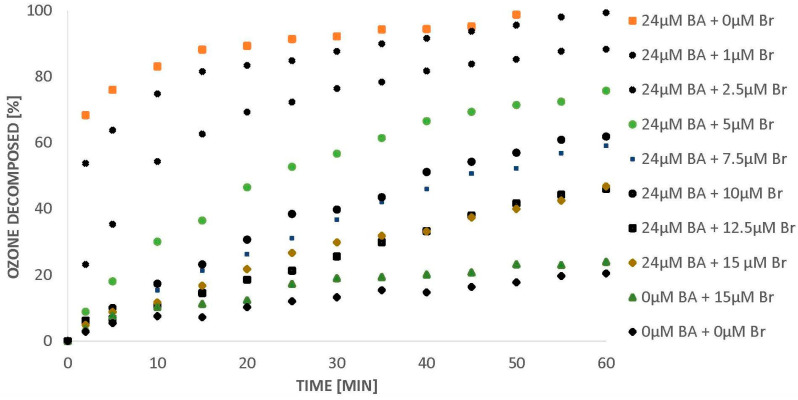
Ozone decomposition in the presence of BA (24 µM) and bromides (0–15 µM), pH 2.5.

**Figure 2 molecules-26-02701-f002:**
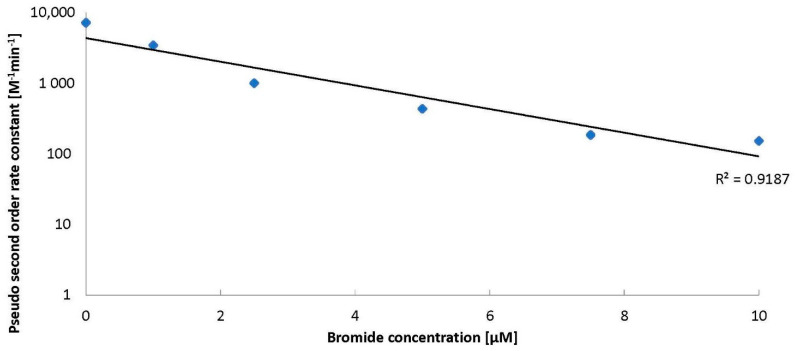
The ozone decomposition kinetics in the presence of BA (24 µM) and bromides (0–10 µM), pH = 2.5.

**Figure 3 molecules-26-02701-f003:**
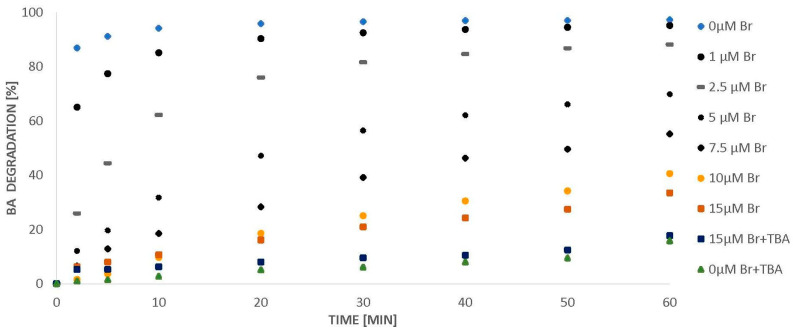
Degradation of BA (24 µM) in the presence of bromides (0–15 µM), pH = 2.5.

**Figure 4 molecules-26-02701-f004:**
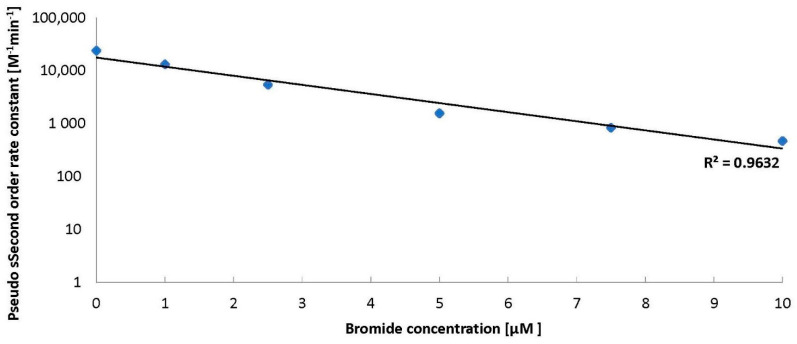
The degradation kinetics of BA (24 µM) in the presence of bromides (0–15 µM), pH = 2.5.

**Figure 5 molecules-26-02701-f005:**
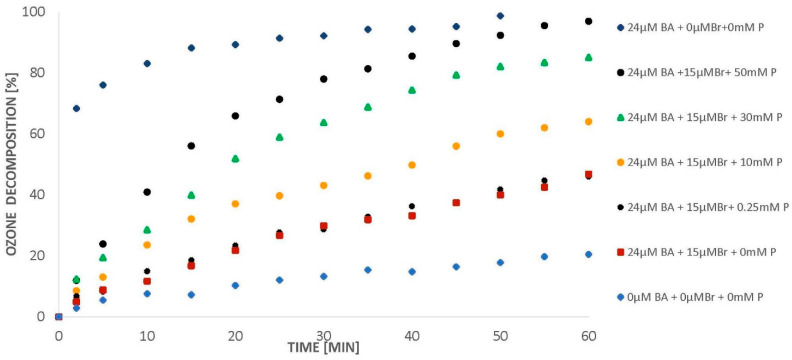
Ozone decomposition in the presence of BA (24 µM), bromides (15 µM), and phosphates (*p* = 0–50 mM), pH = 2.5.

**Figure 6 molecules-26-02701-f006:**
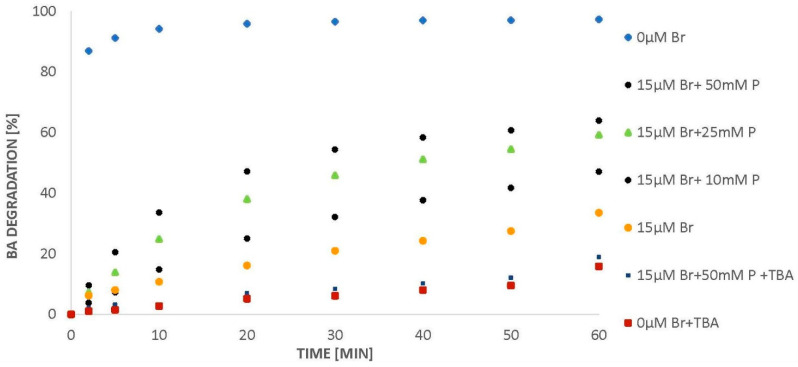
BA degradation (24 µM) in the presence of bromides (15 µM) and a variable amount of phosphates (*p* = 0–50 mM), pH = 2.5.

**Figure 7 molecules-26-02701-f007:**
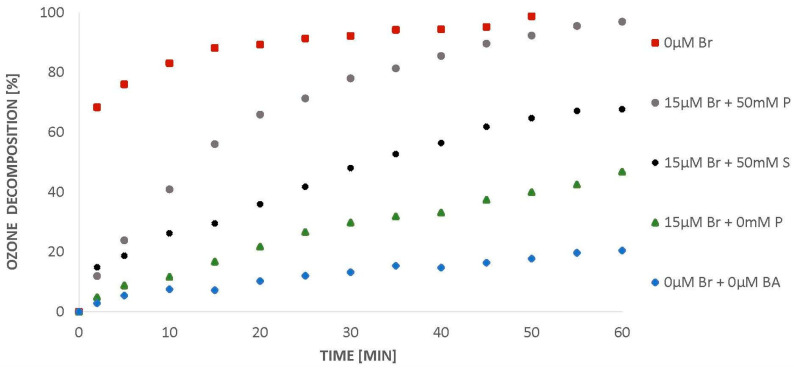
Ozone decomposition in the presence of BA (24 µM), bromides (0 and 15 µM) and 50 mM sulphates (S) or phosphates (P), pH = 2.5.

**Figure 8 molecules-26-02701-f008:**
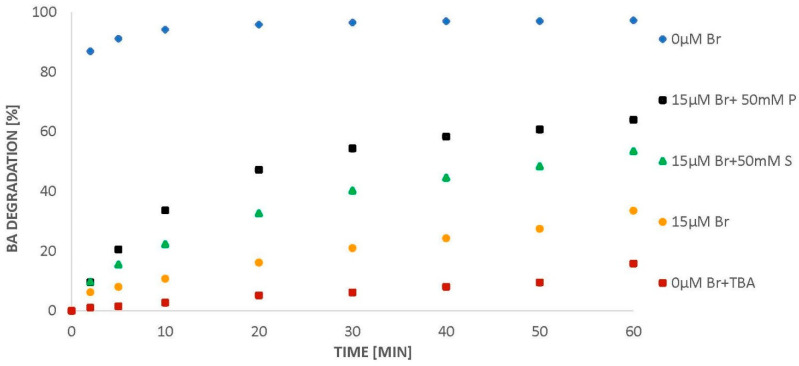
BA degradation (24 µM) in the presence of bromides (0 and 15 µM) and 50 mM sulphates (S) or phosphates (P), pH = 2.5.

**Table 1 molecules-26-02701-t001:** Fraction of hydroxyl radicals (%) reacted with benzoic acid (BA = 24 µM), bromides (15 µM) and phosphates (*p* = 50 mM) or sulphates (S = 50 mM).

Process	Processes with Phosphates (P)			Processes with Sulphates (S)		
	BA	P	Bromides	BA	S	Bromides
with P/S	33.7	28.9	37.4	39.1	15.9	44.4
without P/S	47.3	-	52.7	47.3	-	52.7

**Table 2 molecules-26-02701-t002:** Fraction of phosphate/sulphate radicals (%) reacted with benzoic acid (24 µM), bromide (15 µM) and its recombination.

Phosphate Radicals			Sulphate Radicals		
Benzoic Acid	Bromide	Recombination	Benzoic Acid	Bromide	Recombination
9.1	16.2	74.7	33.1	60.2	6.7

## Data Availability

Not applicable.

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
