# Peer review of "The Role of Sulphate and Phosphate Ions in the Recovery of Benzoic Acid Self-Enhanced Ozonation in Water Containing Bromides"

_molecules, 2021, doi:10.3390/molecules26092701_

Round 1

Reviewer 1 Report

The work presented in the manuscript is a natural continuation of the previous study of this research group (ref. 5), where the effects of chloride and phosphate ions were investigated on the degradation of nitrobenzene by ozonation. In this manuscript the role of HO scavenger was played by bromide ions, the self-enhanced ozonation was realized by another aromatic compound, benzoic acid, while, besides phosphate, also sulfate ions were applied to depress the decreasing effect of bromide. The choice of phosphate and sulfate gave a good opportunity for a comparison of the effects of these ions, both of which form oxidative radicals in the reaction with hydroxyl radical. The results have been appropriately interpreted by considering the rate constants of the possible reactions in these systems.    

The manuscript contains interesting results, even if not of cutting edge type. Its structure and composition are suitable, although contains several errors of grammar and some disturbing sentences (e.g., in line 52, in lines 105-105, 287, 345). The use of articles (a/an or the) should be improved (e.g., in lines 47, 186).

Scientifically, the Introduction needs some improvements. The last sentence of the first paragraph is not quite correct because Fenton systems prefer low pHs for the efficient generation of hydroxyl radicals. The second sentence of the 2nd paragraph should be completed with at least one citation. The last sentence of the Introduction (lines 73-75) is not logical on the basis of the previous parts because no piece of information is given regarding either phosphate or sulfate ions in these systems. Some relevant results and features of them ought to be inserted into the Introduction before choosing them for investigation.  

In Figures 2 and 4, in the titles of the y-axis -1 ought to be superscript. In some figure and table captions, there are superfluous commas or full points. In lines 184-185, citations are given as superscripts (not in brackets) indicating the wrong adaption of the original text.

In lines 191-192, “reaction order” is a sloppy phrase.

The rate constant for Equation (6) is given at pH=7, while the other (for Eq 5) at pH=4. Since the experiments were carried out at low pH, the rate constants should have been known for such circumstances, especially if they are pH-sensitive.

The structure of Table 2 is rather fuzzy. It should be more unambiguous.

Author Response

Reviewer 1

The work presented in the manuscript is a natural continuation of the previous study of this research group (ref. 5), where the effects of chloride and phosphate ions were investigated on the degradation of nitrobenzene by ozonation. In this manuscript the role of HO scavenger was played by bromide ions, the self-enhanced ozonation was realized by another aromatic compound, benzoic acid, while, besides phosphate, also sulfate ions were applied to depress the decreasing effect of bromide. The choice of phosphate and sulfate gave a good opportunity for a comparison of the effects of these ions, both of which form oxidative radicals in the reaction with hydroxyl radical. The results have been appropriately interpreted by considering the rate constants of the possible reactions in these systems.    

The manuscript contains interesting results, even if not of cutting edge type. Its structure and composition are suitable, although contains several errors of grammar and some disturbing sentences (e.g., in line 52, in lines 105-105, 287, 345). The use of articles (a/an or the) should be improved (e.g., in lines 47, 186).

We have improved language and sentences in all content of the manuscript body

Scientifically, the Introduction needs some improvements. The last sentence of the first paragraph is not quite correct because Fenton systems prefer low pHs for the efficient generation of hydroxyl radicals. 

Text has been rewritten according to reviewer’s suggestions. We have added the Fenton process as an example of AOP that operated at low pH. Lines 33-34

The second sentence of the 2nd paragraph should be completed with at least one citation.

Text has been rewritten and extended with additional citations  (line 41) according to reviewer’s suggestions

The last sentence of the Introduction (lines 73-75) is not logical on the basis of the previous parts because no piece of information is given regarding either phosphate or sulfate ions in these systems. Some relevant results and features of them ought to be inserted into the Introduction before choosing them for investigation.

Text has been rewritten and extended with additional explanations according to reviewer’s suggestions. We have added to the introduction (line 77-83)  why bromides, phosphates and sulphates are the objects of the study

In Figures 2 and 4, in the titles of the y-axis -1 ought to be superscript.

We have corrected fig2 and fig3 by making -1 in superscript-. We also explained how the values given on the figures were determined  (lines 120-123) and made precisive the  description of y axis

 trzeba też na osi y dodać pseudo- czyli pseudo second reaction rate constant

 In some figure and table captions, there are superfluous commas or full points.

Thank you for that indication, we have corrected that in all figures and tables captions (figs. 1-8 and tables 1-2).

In lines 184-185, citations are given as superscripts (not in brackets) indicating the wrong adaption of the original text.

Text has been corrected according to reviewer’s suggestions. We have corrected it in lines 214-215

In lines 191-192, “reaction order” is a sloppy phrase.

Text has been rewritten, we have extend that shortcut in lines 221-222

The rate constant for Equation  (6) is given at pH=7, while the other (for Eq 5) at pH=4. Since the experiments were carried out at low pH, the rate constants should have been known for such circumstances, especially if they are pH-sensitive.

Thank you for your comment. Reactions involving phosphates and sulphates at low pH have not been well documented so far. There is no data on this subject in the literature, therefore we decided to include in the paper information about the pH of the presented/cited reactions. We are aware of the significant influence of the pH on the course of these processes. For this purpose, among other, the idea/need for this research was born. Data availability is also one of the reasons for choosing BA as the model compound (we found the most kinetic data)

The structure of Table 2 is rather fuzzy. It should be more unambiguous.

We have notice our oversight and corrected table 2, the  description and values were corrected (table 2)

Reviewer 2 Report

The article is clear and well-writen, but the format needs to be improve. The axes of figures are differents, also the legends are not in good format. Some paragraphs are not in the same format (172-176 lines).

There is no information about:

Author Contributions: line 359
Funding:
Institutional Review Board Statement:
Informed Consent Statement:
Data Availability Statement: line 363
Conflicts of Interest:

Here I have the comments about the article.
The article correctly describes the use of sulfates and phosphates for the stuttering of acidic wastewater containing bromide and makes an important effort to perform the decomposition of benzoic acid molecules. However, there are some doubts that arise throughout the work. References to acidic wastewater, as well as greater importance on the degradation of BA molecules (type of industries in which it is produced, % of wastewater due to this molecule) are lacking in the introduction. During introduction, reactions must be included to better understand of the process. Line 59 mentions seawater, but no reference is given. Ozone decomposition mentions pH 2.5 again, but it is still not adequately described because that pH is selected. The decomposition kinetics associated with the process are of the second order, but it should be referenced or mentioned since on this basis all the work done. Line 104 mentions "paragraph 1" and makes no sense. It is also not mentioned what it is TBA. In Figure 3 and 4 the R2 values obtained are not very high.  Line 156-158 does not understand the process explained well. The reactions of (3) and (4) do not mention products, and should be taken into account in order to better explain the interaction with the other components in the mixture. Tables 1 and 2 are not understood, it would be necessary to  make them again in a clearer way.

Author Response

Reviewer 2

The article is clear and well-written, but the format needs to be improve. The axes of figures are different, also the legends are not in good format. Some paragraphs are not in the same format (172-176 lines).

Thank you for this indication, we have improve formatting of all manuscript body

Author Contributions: line 359
Funding:
Institutional Review Board Statement:
Informed Consent Statement:
Data Availability Statement: line 363
Conflicts of Interest:

Here I have the comments about the article.
The article correctly describes the use of sulfates and phosphates for the stuttering of acidic wastewater containing bromide and makes an important effort to perform the decomposition of benzoic acid molecules. However, there are some doubts that arise throughout the work. References to acidic wastewater, as well as greater importance on the degradation of BA molecules (type of industries in which it is produced, % of wastewater due to this molecule) are lacking in the introduction. During introduction, reactions must be included to better understand of the process. Line 59 mentions seawater, but no reference is given. Ozone decomposition mentions pH 2.5 again, but it is still not adequately described because that pH is selected. The decomposition kinetics associated with the process are of the second order, but it should be referenced or mentioned since on this basis all the work done. Line 104 mentions "paragraph 1" and makes no sense. It is also not mentioned what it is TBA. In Figure 3 and 4 the R2 values obtained are not very high.  Line 156-158 does not understand the process explained well. The reactions of (3) and (4) do not mention products, and should be taken into account in order to better explain the interaction with the other components in the mixture. Tables 1 and 2 are not understood, it would be necessary to  make them again in a clearer way.

References to acidic wastewater, as well as greater importance on the degradation of BA molecules (type of industries in which it is produced, % of wastewater due to this molecule) are lacking in the introduction

Text has been rewritten and extended with additional explanations according to reviewer’s suggestions. We have added anthropogenic sources of BA in the environment (lines 87-92) and explained why benzoic acid  was chosen as a model compound in this study (lines 93-97)

During introduction, reactions must be included to better understand of the process.

Thank you for your comment. Process of ozone decomposition in water is very complex and cannot be described by simple equations, therefore we do not introduce the reactions. In line 37-38 information about ozone stability in different pH has been added. The OH ions are one of its decomposition initiators.

The work focused on so called self-enhanced ozonation reaction. Authors (Huang et al 2015) suggested that in this process, intermediated products of  benzoic acid oxidation reaction enhanced ozone decomposition and rise efficiency of BA degradation. Since self-enhanced reaction, as well as our observations, are quite new reports they are not well known and documented. Therefore, further research are needed to better understand these processes.

Line 59 mentions seawater, but no reference is given

Thank you for your comment. The information about bromides content in different types of water were taken from ref [9] line 65

Ozone decomposition mentions pH 2.5 again, but it is still not adequately described because that pH is selected

Thank you for the comment. Self-enhanced ozonation has been documented primarily as an acidic pH reaction (Huang 2015). As this is a fairly new concept, there are still many aspects to be explored. We decided not to introduce another variable (pH) to the process and use the information that is available in the literature for pH = 2.5. For a brief explanation, see lines 93-97.

The decomposition kinetics associated with the process are of the second order, but it should be referenced or mentioned since on this basis all the work done

Text has been rewritten and extended with additional explanations according to reviewer’s suggestions. We have added in the lines 120-123,the explanation why in this work we operate (pseudo)second order reactions rate constants.

Line 104 mentions "paragraph 1"

Text has been corrected according to reviewer’s suggestions. We have corrected paragraph 1 to paragraph 2.1, line 130

It is also not mentioned what it is TBA

We have extend the abbreviation of TBA in lines 254-255. This information is also given in section 3 lines 306-307

In Figure 3 and 4 the R2 values obtained are not very high.

Thank you for your comment. The presented results are completely new and require further research for better understanding. The obtained correlations indicate the occurrence of some reactions that could have taken place in the solution but are still unknown. Nevertheless, the relatively high value of R2 indicates that bromides are the main factor responsible for the lower efficiency of ozone decomposition and BA degradation.

Line 156-158 does not understand the process explained well.

Text has been rewritten and extended with additional explanations according to reviewer’s suggestions . The explanation is given in lines 184-186

The reactions of (3) and (4) do not mention products, and should be taken into account in order to better explain the interaction with the other components in the mixture.

Thank you for your comment. It is true that knowing all the reaction products allows us to explain the processes  which take place in a solution. However, the phenomena’s described in this work are not well documented yet. It is speculated, in the literature, that phosphate reacts with aromatic compounds by substitution and then the products of these reactions are hydrolyzed. Oxidation with sulphate radicals can occur by direct electron transfer or, as in the case of phosphates, by addition. However, no specific evidence has been given about the reaction pathway, therefore the authors of the cited studies also do not provide a complete list of reaction products. Further research are needed to better understand the observed phenomena’s

Tables 1 and 2 are not understood, it would be necessary to  make them again in a clearer way.

Tables 1 and 2 have been corrected according to reviewer’s suggestions. New descriptions was added. We also notice our oversight and corrected the values according to table descriptions (table 2)